

# Effect of chronic alcohol consumption on oral microbiota in rats with periodontitis

Zirui Zhao[1], Xiao Zhang[1], Wanqing Zhao[1], Jianing Wang[1], Yanhui Peng[1], Xuanning Liu[1], Na Liu[2] and Qing Liu[1]

[1] Hebei Key Laboratory of Stomatology, Hebei Clinical Research Center for Oral Diseases, School and Hospital of Stomatology, Hebei Medical University, Shijiazhuang, Hebei, China
[2] Department of Preventive Dentistry, School and Hospital of Stomatology, Hebei Medical University, Shijiazhuang, Hebei, China

## ABSTRACT

**Background:** The imbalance of oral microbiota can contribute to various oral disorders and potentially impact general health. Chronic alcohol consumption beyond a certain threshold has been implicated in influencing both the onset and progression of periodontitis. However, the mechanism by which chronic alcohol consumption affects periodontitis and its association with changes in the oral microbial community remains unclear.

**Objective:** This study used 16S rRNA gene amplicon sequencing to examine the dynamic changes in the oral microbial community of rats with periodontitis influenced by chronic alcohol consumption.

**Methods:** Twenty-four male Wistar rats were randomly allocated to either a periodontitis (P) or periodontitis + alcohol (PA) group. The PA group had unrestricted access to alcohol for 10 weeks, while the P group had access to water only. Four weeks later, both groups developed periodontitis. After 10 weeks, serum levels of alanine aminotransferase and aspartate aminotransferase in the rats' serum were measured. The oral swabs were obtained from rats, and 16S rRNA gene sequencing was conducted. Alveolar bone status was assessed using hematoxylin and eosin staining and micro-computed tomography.

**Results:** Rats in the PA group exhibited more severe periodontal tissue damage compared to those in the periodontitis group. Although oral microbial diversity remained stable, the relative abundance of certain microbial communities differed significantly between the two groups. *Actinobacteriota* and *Desulfobacterota* were more prevalent at the phylum level in the PA group. At the genus level, *Cutibacterium, Tissierella, Romboutsia, Actinomyces, Lawsonella, Anaerococcus,* and *Clostridium_sensu_stricto_1* were significantly more abundant in the PA group, while *Haemophilus* was significantly less abundant. Additionally, functional prediction using Tax4Fun revealed a significant enrichment of carbohydrate metabolism in the PA group.

**Conclusion:** Chronic alcohol consumption exacerbated periodontitis in rats and influenced the composition and functional characteristics of their oral microbiota, as indicated by 16S rRNA gene sequencing results. These microbial alterations may contribute to the exacerbation of periodontitis in rats due to chronic alcohol consumption.

Corresponding authors
Na Liu, liuna@hebmu.edu.cn
Qing Liu, liuqing@hebmu.edu.cn

# INTRODUCTION

Alcohol consumption represents a significant global health concern and stands as a major risk factor contributing to the global burden of disease (*Kulbida et al., 2024*). It ranks as the seventh leading cause of death and disability worldwide. Studies indicate that exceeding recommended alcohol intake levels (*Mukamal, 2020*) can lead to various health complications, including an elevated risk of conditions such as liver cirrhosis (*Roerecke et al., 2019*), cardiovascular disease (*Roerecke, 2021*), oral cancer and gastrointestinal cancer (*Yoo et al., 2022*).

Periodontitis is a prevalent chronic non-transmissible disease with an escalating incidence annually (*Nazir et al., 2020*; *Trindade et al., 2023*). It manifests as microbial infection of periodontal tissues due to oral microbial dysbiosis, culminating in chronic inflammatory degradation of periodontal tissues and, in severe cases, premature tooth loss (*Papapanou et al., 2018*).

Bacteria inhabiting and proliferating within the periodontal pockets of individuals with periodontitis continually disseminate into the oral cavity, with the oral microbiota serving as a primary source for the recolonization of periodontal pathogens beneath the gum (*Sedghi et al., 2021*). Alterations in the oral microbial community serve as pivotal risk factors in the onset and progression of periodontitis. Emerging research suggests that excessive alcohol consumption could influence the composition of the oral microbial community in humans, potentially depleting beneficial commensal bacteria while fostering the colonization of pathogenic strains (*Fan et al., 2018*). The oral microbiota is subject to influences from various factors such as genetics, environment, and behavior (*Baker et al., 2023*). Chronic alcohol consumption, for instance, may significantly disrupt the homeostasis between oral bacteria and host responses. To date, there is a paucity of studies examining alterations in the oral microbial community within periodontitis rat models subjected to chronic alcohol consumption. This study aims to assess the impact of chronic alcohol consumption on periodontitis through the lens of oral microbiota. We collected oral microbiota samples from two groups of rats and investigated the species composition and functional characteristics of the oral microbiota in rats exhibiting chronic alcohol consumption alongside periodontitis using high-throughput 16S rRNA gene sequencing technology. We aim to uncover potential associations between alterations in oral microbiota induced by alcohol consumption and those resulting from periodontitis, as well as to elucidate the potential role of oral microbiota in mediating the influence of chronic alcohol consumption on periodontitis.

# MATERIALS AND METHODS

## Experimental animals

Twenty-four male SPF Wistar rats (8 weeks old, weighing 200–220 g) were obtained from Beijing Hfk Bioscience Co., Ltd. (License number: SCXK (Jing) 2019-0008). They were

raised at the Hebei Medical University Laboratory Animal Public Service Platform. The study was approved by the Ethics Committee of the Hospital of Stomatology Hebei Medical University (Approval number: [2022]001) and adhered to the ARRIVE guideline. Briefly, the rats were housed in pairs (two animals/cage) at a temperature of 20–25 °C and a relative humidity of 40–70%, with a 12-h light/dark cycle (lights on at 08:00 a.m.). They had *ad libitum* access to rodent chow and water. After a week of acclimatization, the experiment was initiated.

## Experimental design

After 1 week of adaptation, the rats were randomly allocated into two groups of twelve each using a random number table. The experimental groups were as follows: Periodontitis group (P group) and periodontitis + alcohol group (PA group).

The P group had free access to water, while the PA group received an adaptive regimen of 10% (v/v), 15% (v/v), 20% (v/v), and 25% (v/v) alcohol solutions in their drinking water over 16 days, followed by maintenance feeding of 30% (v/v) alcohol in drinking water for 10 weeks, as per the established protocols (*de Almeida et al., 2020*; *Surkin et al., 2014*). The alcohol solution, prepared by diluting ethanol (GR: Guaranteed reagent) in distilled water, served as the sole liquid source for rats in the PA group, with each rat consuming approximately 15–20 ml per day at unspecified times.

## Establishment of periodontitis model

Periodontitis induction commenced 4 weeks after initiating the 30% (v/v) alcohol-solution regimen. Rats were weighed, and their lower abdomen was disinfected. Intraperitoneal injection of 0.6% pentobarbital sodium (6.7 mL/kg) anesthetized the rats. The gingival sulcus of the maxillary first molar was separated with a dental probe, and a 0.25-mm diameter orthodontic ligature wire and cotton thread were threaded through the interdental gap between the maxillary first and second molars. Any excess ligature was trimmed to prevent oral activity restriction. Ligature status was monitored weekly, with re-fixation or re-ligation performed as necessary.

## Specimen collection and processing

At the end of the tenth week, periodontal tissue health assessment was conducted. Rats were fasted for 12 h, anesthetized with isoflurane, and blood was collected *via* the retroorbital plexus. Serum was obtained by centrifugation at 13,000 r/min for 10 min in a low-temperature centrifuge, and the upper serum was aspirated and stored at −80 °C. Oral swabs were collected from the soft cavity and hard tissue surfaces of the rats in the following order: tongue surface, cheek, palate, maxillary molars, tongue ventral side and floor of the mouth, lower incisors and mandibular vestibule, upper incisors and maxillary vestibule, with constant attention to prevent contamination. The swabs were promptly frozen in liquid nitrogen before storage at −80. Finally, the rats were euthanized with an overdose of 0.6% pentobarbital sodium, and maxillary alveolar bone tissue was collected and fixed in 4% paraformaldehyde.

## Biochemical analysis

Levels of alanine aminotransferase (ALT) and aspartate aminotransferase (AST) in blood samples were determined using a fully automatic biochemical analyzer (Mindray, Shenzhen, China).

## Hematoxylin-eosin staining

Maxillary bone tissue was fixed in 4% paraformaldehyde for 48 h, followed by decalcification in a 10% ethylene diamine tetraacetic acid (EDTA) solution at room temperature. Tissue sections (4-μm-thick) from dental and periodontal tissues near the ligation region were sliced along the long axis of the molar tooth near the ligation region. The tissue slices were stained with hematoxylin-eosin (HE) and observed using an Olympus BX63 system under a light microscope.

## Micro-computed tomography analysis to assess the alveolar bone loss

Six rats from each group underwent micro-computed tomography (Micro-CT) scanning of their right maxillary alveolar bone using the Micro-CT system (SkyScan 1276 Bruker, Germany) with scanning parameters set at 49 kV, 200 μA, and a resolution of 10 μm. The three-dimensional reconstruction of the original images was performed using the NRecon program. Alveolar bone loss was assessed by measuring the linear distance between the cemento-enamel junction (CEJ) and the alveolar bone crest (ABC) at mesial, midpoint, and distal sites from the first molar on the buccal and lingual sides of the right maxilla using CT Analyzer.

## 16S rRNA sequencing of rat oral swab

### DNA extraction

The soil and feces genomic DNA extraction kit (Tiangen, Beijing, China) was used to extract bacterial genomic DNA from all oral swab samples using the magnetic bead technique. Subsequently, DNA purity and concentration were assessed using agarose gel electrophoresis.

### 16S rRNA gene amplification

Bio-rad T100 gradient PCR instrument was used to amplify the specific regions of DNA samples, and the corresponding regions of the PCR amplification primers were 16SV3-V4 region primers with Barcode (341F CCTAYGGGRBGCASCAG/806R GGACTACNNGGGTATCTAAT). The enzyme and buffer used were Phusion® high-fidelity PCR master mix with GC buffer (New England Biolabs) to ensure the efficiency and accuracy of amplification. According to the concentration of PCR products, the samples were mixed at the same concentration, after which the PCR products were purified by using the Gene Jet Gel Extraction Kit (Thermo Fisher Scientific, Waltham, MA, USA). The sequence with the main band size of 400–450 bp was selected and tapped to recover the target band.

### Library construction and sequencing

Library construction was performed using the TruSeq DNA PCR-Free Library Preparation Kit (Illumina, San Diego, CA, USA), followed by quantification and detection using Qubit and Q-PCR. After passing quality control, PE250 sequencing was conducted using the NovaSeq 6000 kit at Novegene Co., Ltd. (Beijing, China).

### Bioinformatic analysis

First, data quality control was performed. To get raw tags, the sequencing data was processed using FLASH (Version 1.2.11) for read assembly. Cutadapt was utilized to match the reverse primer sequences and trim residual sequences to ensure downstream analysis accuracy. To obtain clean tags, the raw tags were filtered and processed using the fastp program (Version 0.23.1). These clean tags were aligned and compared to the SILVA database (https://www.arb-silva.de/) for species annotation and chimeric sequence removal, resulting in effective tags. To denoise the effective tags and get the final amplicon sequence variants (ASVs), the QIIME2 program (Version QIIME2-202202) was used, followed by species annotation using the SILVA 138.1 database. The QIIME2 program was also utilized to generate the species evolutionary tree using multiple sequence alignment.

Alpha diversity indices including observed_features, Shannon, Simpson, and chao1, were calculated using the QIIME2 program to evaluate microbial community diversity within samples. Significant variations in species diversity among groups were assessed using the Tukey or Kruskal–Wallis rank sum test, with box plots constructed for visualization. Dilution curves were plotted in R (v4.0.3) to assess sequencing depth.

Beta diversity analysis was conducted using the QIIME2 program to analyze microbial community composition among samples. Principal coordinate analysis (PCoA) and principal component analysis (PCA) were performed using the R program (v4.0.3) to illustrate differences between samples *via* scatter plots.

Statistical analyses, including Anosim, Adonis, T-test, and MetaStat, were performed using R software (v4.0.3) to compare samples between groups and develop box graphs. The LEfSe (LDA Effect Size) statistical analysis was conducted using the LEfSe program, with results displayed as bar charts and evolutionary tree diagrams. Functional annotation of microbial ASV sequences was performed using the Tax4Fun program (V0.3.1), which matched KEGG database prokaryotic whole-genome 16S rRNA gene sequences with the SILVA SSU Ref NR database.

## General statistical analysis of samples

General experimental data was statistically analyzed using SPSS 26. Quantitative data were presented as mean ± standard deviation (X ± S). Before statistical tests, multiple samples' normality and variance homogeneity were confirmed. For normally distributed and homogenous data, a T-test was used; for normally distributed data with an uneven variance, Welch's t-test was used; for non-normally distributed data, a non-parametric Mann-Whitney U-Test was used ($\alpha = 0.05$, $P < 0.05$).

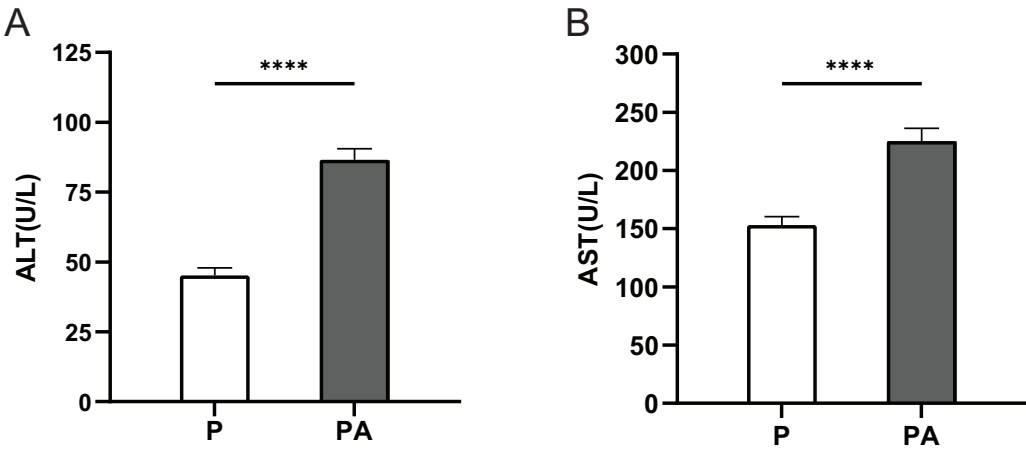

**Figure 1  The expression of ALT and AST in rats of each group.** (A) ALT; (B) AST. $^{****}P < 0.0001$.

## RESULTS

### Serum ALT and AST levels

The serum ALT and AST levels were significantly elevated in the PA group compared to the P group ($P < 0.0001$) (Fig. 1). This indicates that after 10 weeks of alcohol consumption, rats in the PA group developed liver damage, demonstrating the adequacy of chronic alcohol consumption in the PA group.

### Histopathological observation

In the P group, attachment loss was observed at the CEJ, accompanied by degradation of periodontal ligament fibers and minor infiltration of inflammatory cells. Conversely, the PA group exhibited more severe periodontal tissue deterioration, characterized by collagen and periodontal ligament fibers, migration of attachment epithelium toward the root, and extensive infiltration of inflammatory cells in the lamina propria and epithelium (Fig. 2). These findings suggest a greater extent of periodontal destruction due to alcohol consumption.

### Micro-CT analysis

Micro-CT analysis of the rats' right maxilla confirmed significant evidence of periodontitis and alveolar bone resorption in both groups. The CEJ-ABC distance was significantly greater in the PA group compared to the P group ($P < 0.01$), indicating exacerbation of periodontitis due to chronic alcohol consumption in the rat model (Fig. 3).

### The oral swab 16S amplicon sequencing results of rats
#### Comparison of oral microbial diversity in rats

Analysis of the oral microbiota composition revealed 559 unique ASVs in group P, 1,372 unique ASVs in group PA, and 387 common ASVs in both groups. The species dilution curves for both groups flattened with increasing sequencing data, indicating that the sequencing data was adequate for diversity analysis. More data would not have

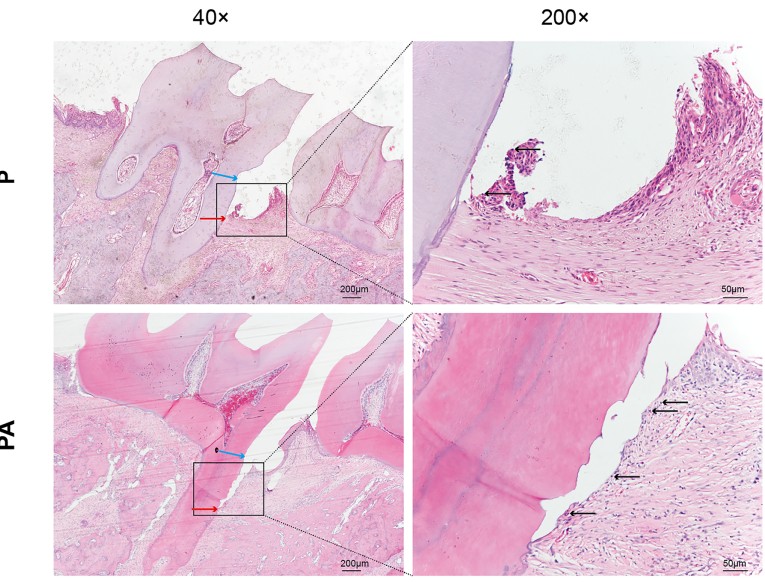

**Figure 2 Distal periodontal tissue of mandibular first molar in two groups of rats.** Staining: hematoxylin and eosin (HE). Blue arrows show cemento–enamel junction (CEJ). Red arrows show the coronalside of junctional epithelium. Black arrows show inflammatory cell infiltration in the lamina propria and epithelium (arrows).

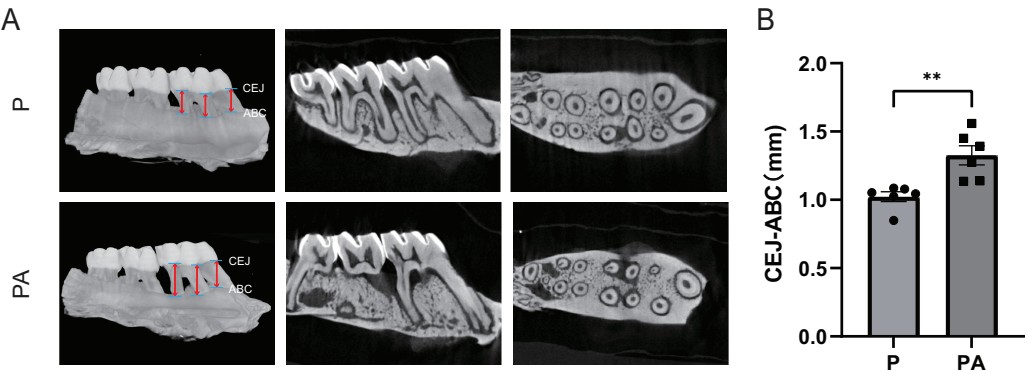

**Figure 3 Micro-CT scanning of rats' right alveolar bone.** (A) Micro-CT image; (B) CEJ-ABC distance. Red arrows show the points for measuring linear alveolar bone loss, taken at mesial, midpoint, and distal sites around the first molar, both on the buccal and lingual sides. CEJ, cemento–enamel junction; ABC, alveolar bone crest. ${}^{**}P < 0.01$.

significantly influenced the alpha diversity index, and the sequencing data was adequate to reflect the majority of microbial diversity in the samples and the actual state of the oral microbiota in the rats (Fig. 4). The alpha diversity indicators of community diversity are described using chao1, observed_otus, Shannon, and Simpson indices. The absence of statistically significant differences in the alpha indices between the two groups indicated that chronic alcohol consumption did not impact the richness and community diversity of the oral microbiota in rats with periodontitis (Fig. 5).

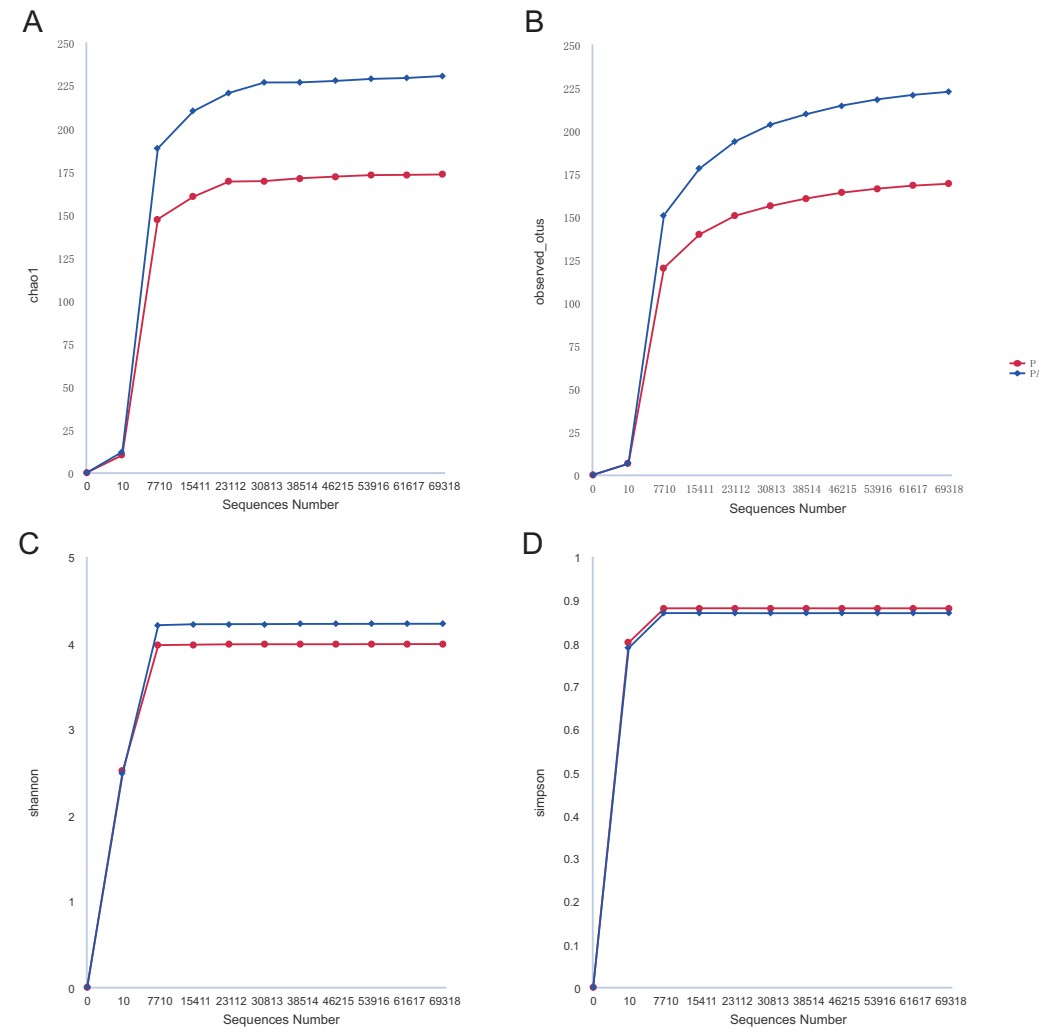

**Figure 4 Rarefaction curves of alpha diversity.** (A) Chao1; (B) observed_otus; (C) Shannon; (D) Simpson.

***Chronic alcohol consumption modified the composition of the oral microbiota at multiple taxonomic levels in rats with periodontitis***

To assess the differences in microbial community composition between the two groups, Beta diversity PCoA based on weighted and unweighted UniFrac distances was performed (Fig. 6). Analysis categorized oral microbial communities into two groups, indicating significant differences in community composition between the two groups. The Anosim and Adonis indices further confirmed significant variations in microbial community composition between the groups (Table 1).

To investigate changes in oral microbiota, ASVs were aggregated at multiple taxonomic levels to compare species' relative abundance and occurrence. Based on the species annotation findings, the top ten species with the highest relative abundance at the phylum and genus levels were selected to construct a relative abundance histogram for each group (Fig. 7).

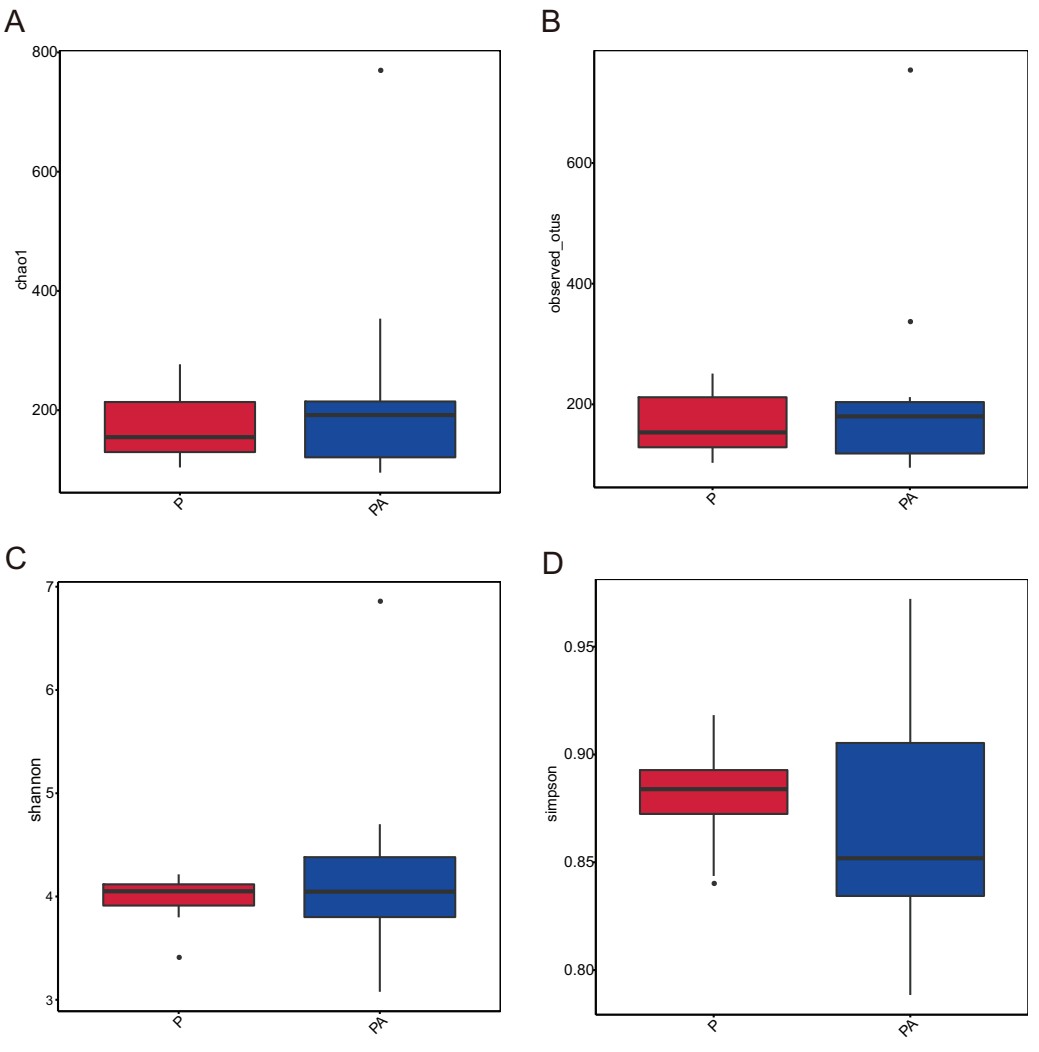

**Figure 5 Box plot of alpha diversity showing differences between groups.** (A) Chao1; (B) observed_otus; (C) Shannon; (D) Simpson.

At the phylum level, the oral microbiota in the two groups was mostly composed of four phyla: *Proteobacteria*, *Firmicutes*, *Bacteroidota*, and *Actinobacteriota*. Further comparison of inter-group differences in the composition of microbiota in samples using MetaStat revealed that the PA group had a significantly higher relative abundance of *Actinobacteriota* (6.12% *vs.* 2.44%) and *Desulfobacterota* (0.37% *vs.* 0.03%) compared to the P group (Fig. 8). *Bacteroidota's* relative abundance increased (9.80% *vs.* 4.16%), whereas *Firmicutes'* relative abundance declined (17.69% *vs.* 20.01%), however, the change was not statistically significant.

At the genus level, compared to the P group, the relative abundance of *Cutibacterium* (0.48% *vs.* 0%), *Tissierella* (0.40% *vs.* 0%), *Romboutsia* (0.18% *vs.* 0.03%), *Actinomyces* (0.15% *vs.* 0.01%), *Lawsonella* (0.08% *vs.* 0%), *Anaerococcus* (0.07% *vs.* 0%), and *Clostridium_sensu_stricto_1* (0.07% *vs.* 0%) increased significantly in the PA group, while the abundance of *Haemophilus* (0.08% *vs.* 1.50%) significantly declined (Fig. 9). The

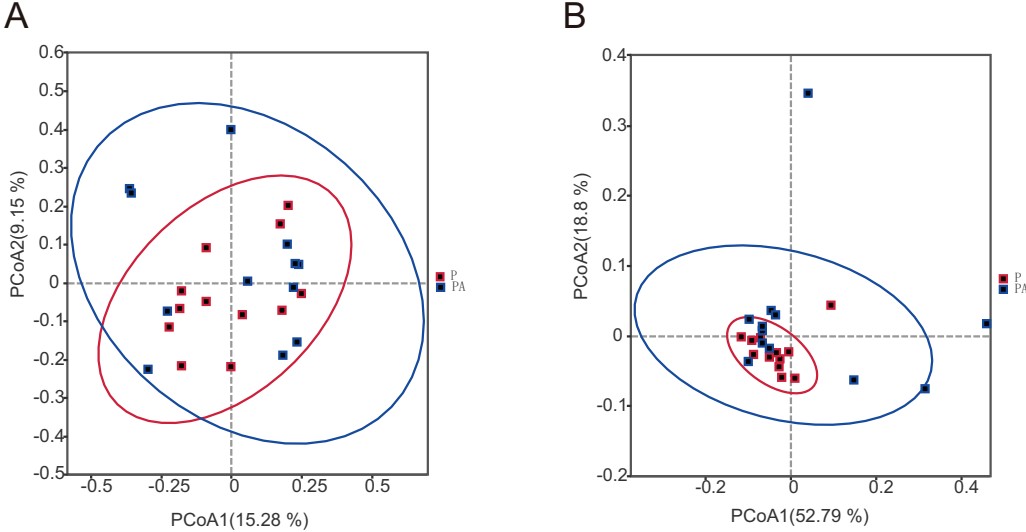

**Figure 6 The PCoA diagram shows that the composition of oral microbial community is different between groups.** (A) Based on unweighted UniFrac distance; (B) based on weighted UniFrac distance.

**Table 1 Differences between groups were analyzed by Anosim and Adonis.**

**Anosim**

| Group | R-value | P-value |
|---|---|---|
| P-AP | 0.125 | 0.01* |

**Adonis**

| Group | SumsOfSqs | MeanSqs | F.Model | R2 | Pr(>F) |
|---|---|---|---|---|---|
| P-AP | 0.3135 | 0.3135 | 2.13905 | 0.08865 | 0.01* |

Note:
* $P < 0.05$

relative abundance of *Ligilactobacillus* (0.60% *vs.* 0.08%), *Pseudomonas* (0.63% *vs.* 0.20%), and *Prevotellaceae*_UCG-001 (0.28% *vs.* 0.03%) significantly increased, while *Veillonella* (6.82% *vs.* 12.81%) and *Acinetobacter* (1.21% *vs.* 2.67%) dropped, but the differences were not statistically significant.

LEfSe (LDA Effect Size) analysis identified statistically distinct species and their influence between groups. In addition, an evolutionary branch diagram (Fig. 10) was generated to illustrate the dominant species at different taxonomic levels from phylum to species.

The LEfSe analysis revealed that the relative abundance of *Propionibacteriales, Propionibacteriaceae, Cutibacterium, Lachnospirales, Lachnospiraceae, Peptostreptococcaceae, Family_XI, Chitinophagales, Chitinophagaceae, Clostridia*, and *Peptostreptococcales_Tissierellales* were significantly higher in the PA group than the *P* group, while *Moraxellaceae, Acinetobacter, Haemophilus, Streptococcus_mutans*, and *Sphingobacterium* were significantly lower.

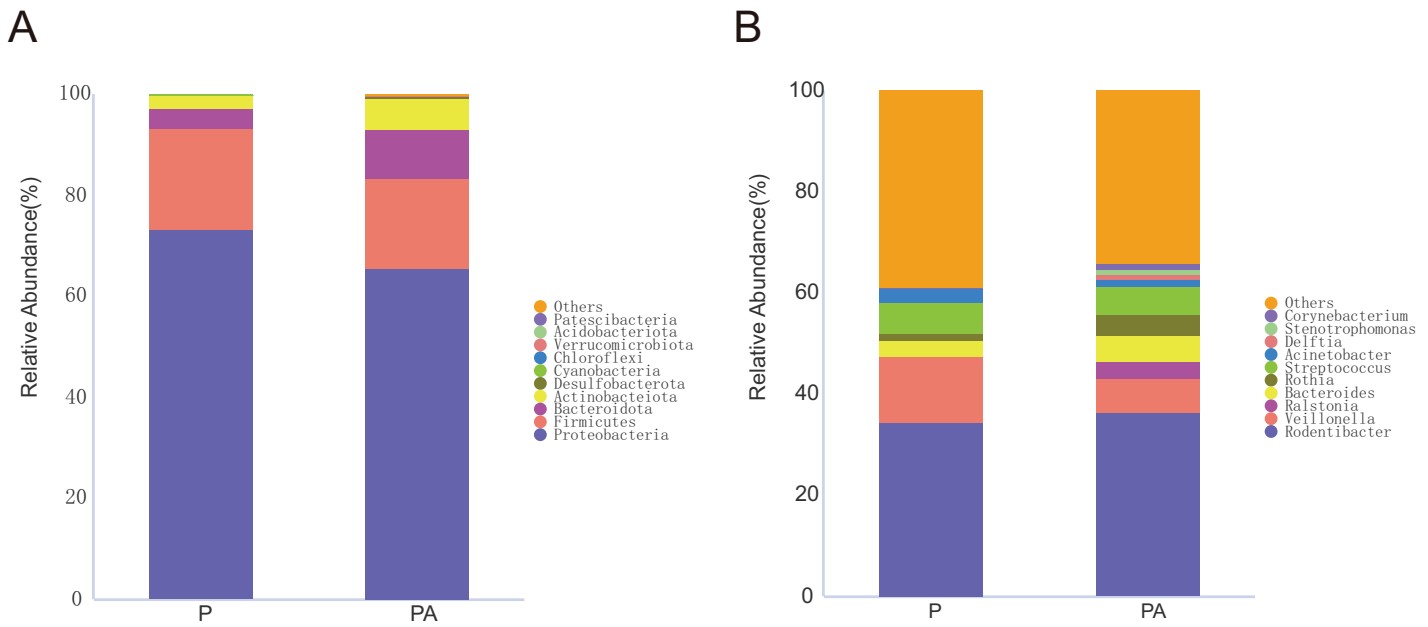

**Figure 7 Column diagram of relative abundance of oral flora species at multi-classification level in two groups, Only the top 10 species in relative abundance are shown in the figure.** (A) The phylum level, (B) the genus level.

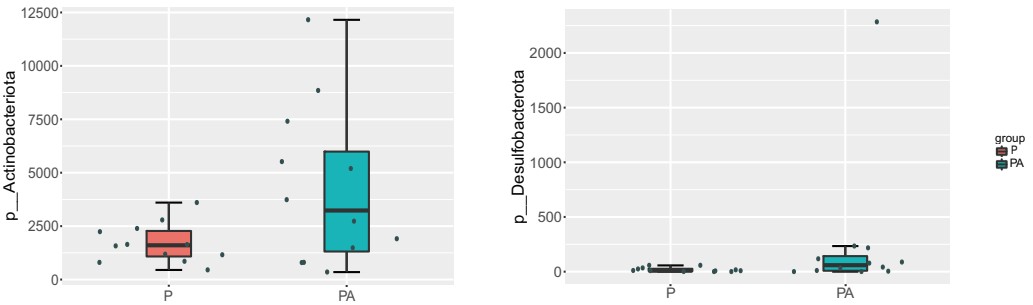

**Figure 8 Box diagram of species difference analysis between groups at the phylum level.**

### *The functional characteristics of oral microbiota predicted using Tax4Fun*

Tax4Fun function exhibits superior gene prediction accuracy of rat oral microbiota compared to PICRUSt function prediction, reflecting the functional characteristics of the microbiota to some extent. The functional characteristics of oral microbiota primarily involved environmental information processing, genetic information processing, metabolism, organic systems, human diseases, and cellular processes, with the PA group significantly outperforming the *P* group in carbohydrate metabolism function (Fig. 11).

## DISCUSSION

Chronic alcohol consumption poses significant risks to the host, as alcohol's lingering effects on the body can lead to various adverse health outcomes. Direct oral impacts of alcohol include associations with oral and pharyngeal cancers, dental caries, and

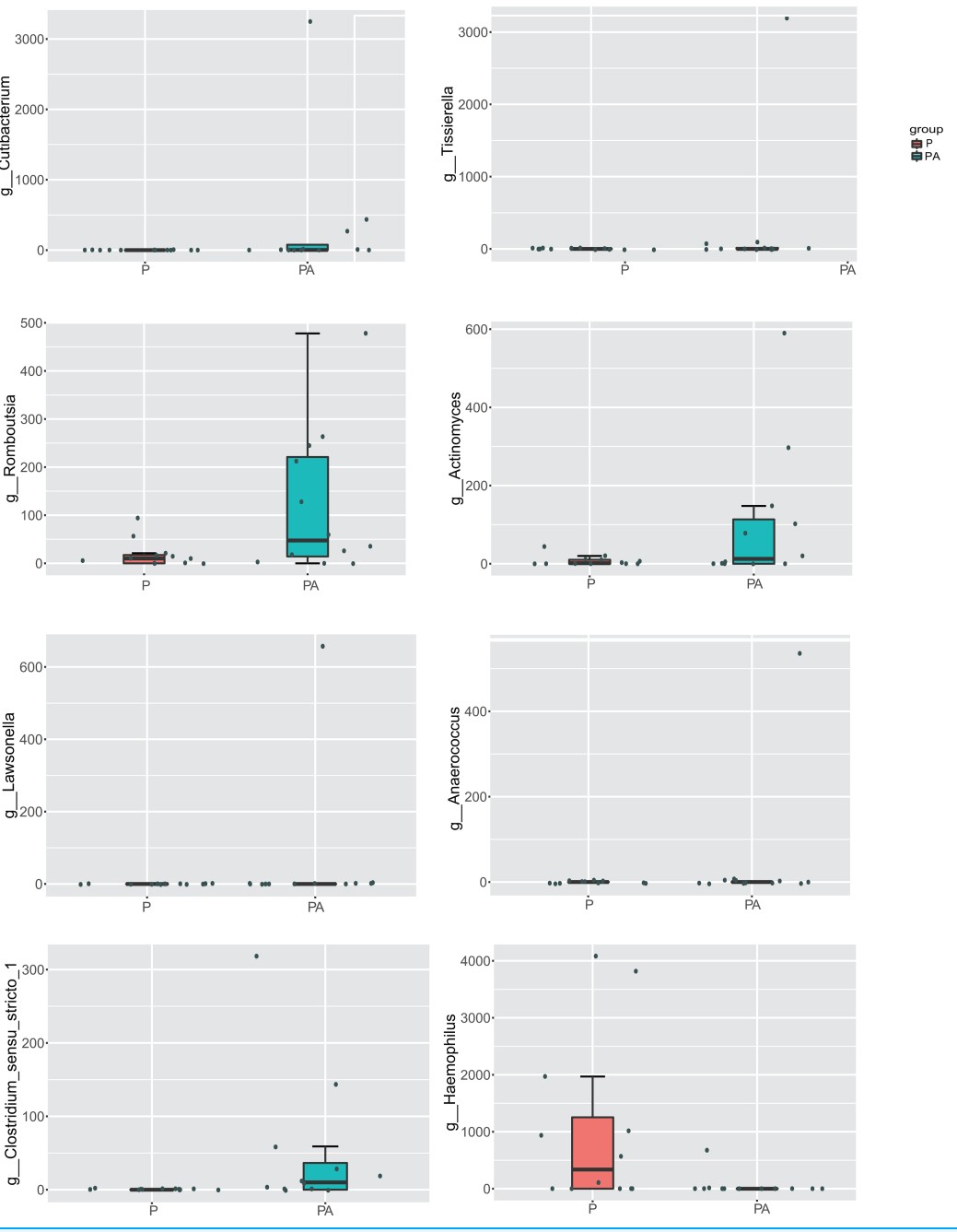

**Figure 9 Box diagram of species difference analysis between groups at the genus level.**

periodontal disease. Periodontitis is closely associated with subgingival bacteria, and investigations into subgingival bacterial communities often correlate with periodontal diseases. However, it is worth noting that periodontal health is influenced by microbial activity in other oral cavity regions, such as the tongue, cheeks, and palate (*Belstrøm et al., 2021*). While subgingival plaque analysis remains a key aspect in understanding periodontal status, alterations in oral plaque bacteria can also provide valuable insight into
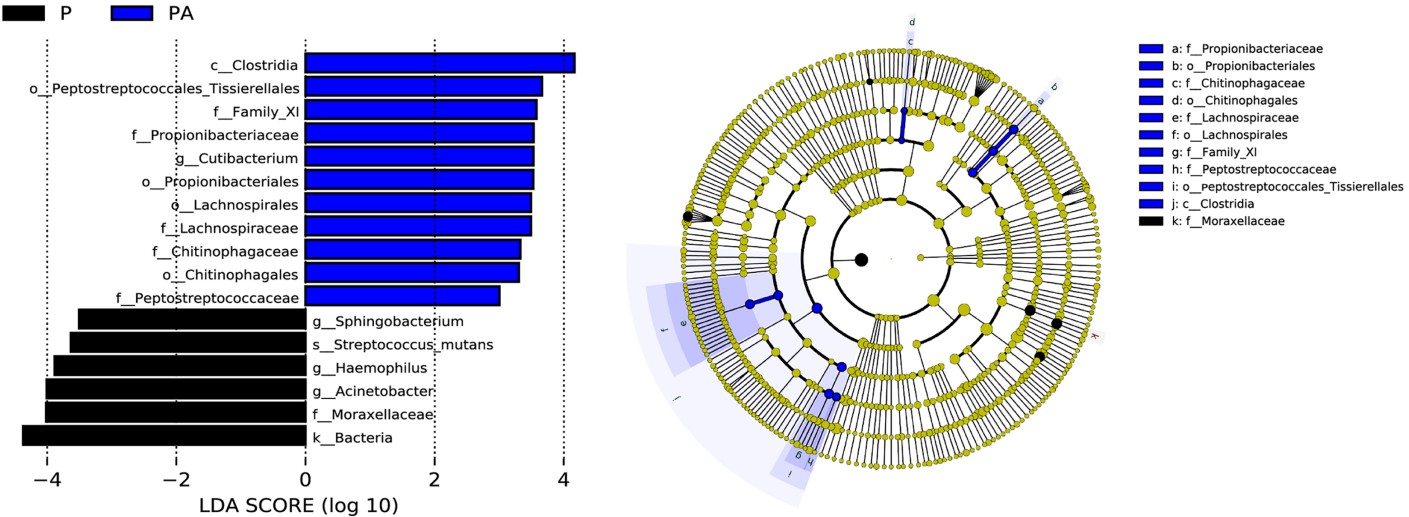

**Figure 10 LEfSe analysis shows the differences in the composition of oral microflora between groups, only the species with LDA > 3.0 are displayed.** (A) The histogram of LDA distribution; (B) the evolutionary branching diagram.  

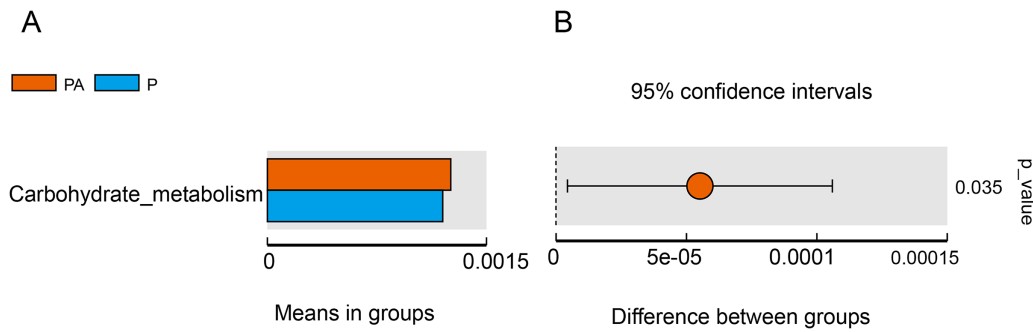

**Figure 11 Difference function between groups.** (A) Each bar represents the average abundance of difference function between different groups; (B) the confidence level of the difference between groups.

periodontitis. After the gut microbial community, the oral microbial community has the highest microbial content in the human body. Current research suggests that the oral microbial community is relatively stable and resilient compared to the gut microbial, with factors like diet and antibiotics exerting minimal influence on oral microbial composition (*Tuganbaev, Yoshida & Honda, 2022*).

Numerous studies have established the influence of chronic alcohol consumption on the onset and progression of periodontitis. Consistent with previous findings, our study revealed that chronic alcohol consumption over 10 weeks, with a 6-week the first molar ligations, led to more severe bone loss and histological destruction in the PA group compared to the P group (*de Almeida et al., 2020*; *Hamdi et al., 2021*).

Our investigation further explored the impact of chronic alcohol consumption on oral microbial communities in periodontitis. Despite expectations, we found significant differences in the chao1, observed_features, Shannon, and Simpson indices between the PA and P groups. This suggests that the number of detected species was similar across both

groups and that inoculation with *Porphyromonas gingivalis* had no effect on microbial community diversity (*Walkenhorst et al., 2020*). Similar studies have also shown that persistent alcohol consumption for 20 weeks did not alter gut microbiota diversity (*Frausto et al., 2022*). However, contrasting findings from other studies suggest that alcohol consumption may reduce alpha diversity levels and diminish bacterial variety (*Lin et al., 2020*). Higher alpha diversity in microbial communities typically indicates an abundance of symbiotic bacteria that aids in disease prevention and bodily homeostasis. Previous research has indicated that the P group exhibits a higher oral microbial community diversity compared to healthy control groups (*Ortiz et al., 2022*; *Zhang et al., 2021*). Moreover, chronic consumption of alcohol can also contribute to gut bacterial overgrowth and dysbiosis. Our divergent results could be attributed to differences in modeling methodologies and sample sites. While alpha diversity analysis often serves as an indicator of disease status in human health, it has several limitations. Moreover, alterations in diversity do not always signify the presence of a disease (*Fan et al., 2018*).

Variations in the relative abundance of microorganisms were observed between the different groups. Specifically, the PA group exhibited a lower relative abundance of Firmicutes compared to the P group, while showing a higher relative abundance of Bacteroidetes. However, these alterations did not reach statistical significance. Notably, the Firmicutes/Bacteroidetes (F/B) ratio decreased in the P group (3.765 *vs*. 0.228). This ratio has been associated with specific disease states such as obesity (*Magne et al., 2020*), colon cancer (*Flemer et al., 2017*), breast cancer (*An, Kwon & Kim, 2023*), early childhood caries (*Rizzardi et al., 2021*), and Alzheimer's disease (*Vogt et al., 2017*). However, ongoing debate persists regarding the association and mechanisms of action of this biomarker in human disorders, influenced by factors such as sample processing, data analysis methodologies, and confounding factors.

This study revealed a significant enrichment of *Actinobacteria*, *Actinomycetes*, and *Streptococcus* in the PA group. These microorganisms are recognized as opportunistic pathogens that infect the skin, soft tissues, and bones when the host's immune system is compromised (*Sowani et al., 2017*). *Actinomycetes* have been linked to dental plaque biofilm formation in individuals with periodontitis, with certain strains triggering inflammatory cytokines and inducing horizontal alveolar bone loss (*Vielkind et al., 2015*). Additionally, although not statistically significant, *Bacteroidetes* abundance doubled in the PA group. *Bacteroidetes*, comprising diverse Gram-negative bacteria, are known for their outer membrane component, lipopolysaccharides (LPS), which can promote the release of inflammatory factors and induce inflammation (*Yussof et al., 2020*). Furthermore, other bacteria associated with periodontitis showed increased abundance in the PA group, including *Tissierella*, *Anaerococcus*, *Lawsonella*, *Capnocytophaga*, *Pseudomonas* and *Clostridia*, most of which are Gram-negative anaerobic or microaerobic bacteria. This suggests that alcohol consumption may create an anaerobic environment in the oral cavity of rats with periodontitis, facilitating the growth of anaerobic microbes that can lead to various infections in the human body, including lung abscesses, brain abscesses, postoperative infections, and oral inflammation (*Bell et al., 2016*; *Caufield et al., 2015*; *Cobo, 2022*). These bacteria are also known as periodontal pathogens (*Abusleme et al., 2013*;

*Ganly et al., 2019*), with increased content observed in the oral microbial community of patients with periodontal disease (*Di Stefano et al., 2022*).

*Desulfarculales* and *Desulfarculaceae* were significantly enriched in the PA group. *Desulfarculaceae*, Gram-negative anaerobic bacteria, produce hydrogen sulfide gas as a final metabolic byproduct, which is toxic to oral epithelial cells, inhibits cytochrome oxidase, affects ATP synthesis, cleaves disulfide bonds in proteins, and inhibits myeloperoxidase and catalase (*Kushkevych et al., 2020*). *Desulfarculaceae* also induces the development of inflammatory cells through LPS or endotoxin, which are outer membrane vesicles (OMV), potentially damaging the host (*Singh, Carroll-Portillo & Lin, 2023*). *Desulfarculaceae* can also induce the pro-inflammatory response of oral epithelial cells (*Cross et al., 2018*). Moreover, this species is abundant in patients with periodontitis (*Heggendorn et al., 2013*; *Liu et al., 2023*).

Similarly, *Lachnospirales* and *Lachnospiraceae* were more abundant in the PA group. This bacterial family is involved in generating short-chain fatty acids and bile acid metabolism, contributing to pathogen resistance and influencing host health. *Lachnospiraceae* abundance has been reported to increase in fecal 16S rRNA analyses of patients with inflammatory bowel illness (*Alam et al., 2020*), colorectal cancer (*Hexun et al., 2023*), and alcoholic liver disease (*Dubinkina et al., 2017*) while decreasing in non-alcoholic cirrhosis (*Bhat et al., 2016*).

In the PA group, there was a notable increase in abundance observed in the order *Propionibacteriales*, family *Propionibacteriaceae*, and genus *Cutibacterium*. *Cutibacterium* is a symbiotic bacterium typically found on healthy skin, particularly near sebaceous glands, and has been implicated in various inflammatory infections, including eye infections, dental pulp disorders, and acne vulgaris (*Kim et al., 2022*). Interestingly, *Cutibacterium* enrichment has been observed in the blood and atherosclerotic plaques of patients with periodontitis (*Isoshima et al., 2021*).

*Peptostreptococcales_Tissierellales*, *Peptostreptococcales_Tissierellales Family_XI*, *Peptostreptococcaceae*, *Peptostreptococcus*, and *Romboutsia* also exhibited higher abundance in the PA group. *Peptostreptococcaceae* encompasses various anaerobic cocci capable of causing infections throughout the body (*Slobodkin, 2014*), with studies indicating a significant increase in *Peptostreptococcales* concentration at periodontitis sites (*Barbagallo et al., 2022*; *Lafaurie et al., 2022*; *Wei et al., 2022*). Additionally, the genus *Romboutsia* is enriched in the gut microbiota of individuals with stage III/B periodontitis (*Kawamoto et al., 2021*).

Conversely, the genus *Haemophilus* exhibited a significant decrease in abundance, consistent with previous findings (*Liao et al., 2022*). *Haemophilus* is considered a symbiotic bacterium in the human body and is part of the healthy core oral bacteriome (*Al-Hebshi et al., 2017*). Moreover, a lower abundance of *Haemophilus* has been associated with diabetes, a significant risk factor for periodontitis with a bidirectional link (*Letchumanan et al., 2022*).

*Lactobacillales*, Gram-positive probiotic bacteria of the phylum Firmicutes, can ferment carbohydrates to produce lactic acid. Long-term ethanol treatment has been shown to reduce the number of Firmicutes and *Lactobacillales* in mice (*Fan et al., 2018*), potentially

promoting the development of alkali-resistant bacteria (*Bull-Otterson et al., 2013*). Moreover, *Lactobacillus rhamnosus GG* treatment lowered intestinal pH and inhibited the proliferation of harmful bacteria in ethanol-fed mice (*Bull-Otterson et al., 2013*). Interestingly, in our investigation, the PA group exhibited a larger relative abundance of usually helpful *Lactobacillus*, yet the difference was not statistically significant. This could be attributed to the fact that *Lactobacillales* can behave as a pathogen under particular circumstances (*Mikucka et al., 2022*), with various *Lactobacillus* triggering varied impacts (*Goldstein, Tyrrell & Citron, 2015*). *Lactobacillus* is generally a healthy and diverse microbial community, but sometimes strains are separated under certain conditions, which has an influence on health (*Pineiro & Stanton, 2007*).

Furthermore, the study evaluated changes in bacterial function, and the "carbohydrate metabolism" function was significantly enhanced in the PA group compared to the P group. Certain carbohydrate metabolic pathways have been associated with periodontitis, such as the butyric acid metabolic pathway (*Sakanaka et al., 2017*) and the ascorbic acid and aldaric acid metabolic pathways (*Qin et al., 2022*). The consumption of alcohol may alter the oral microbiota and, as a result, disrupt local metabolism in the oral cavity, which, in turn, may affect periodontitis in rats. However, it is essential to note that Tax4Fun predicts functional genes solely based on 16S rRNA gene sequencing, and obtaining accurate information about functionality requires metagenomic sequencing.

Several limitations to this study warrant acknowledgment. First, while oral swab samples are convenient to obtain and contain a rich microbiota, they may not accurately represent the microbial composition at specific regions within the oral cavity. For instance, subgingival dental plaque harbors a high concentration of periodontal pathogenic bacteria (*Curtis, Diaz & Van Dyke, 2020*). Therefore, future research should consider collecting samples from particular oral locations to better capture localized microbial variations. Additionally, while this study demonstrated significant changes in the composition of numerous bacteria in periodontitis, further research is required to validate if these changes contribute to periodontitis development. Second, while the alcohol in the drinking water model is physiological, inexpensive, and can be manipulated to simulate human drinking patterns, challenges arise in precisely regulating and maximizing alcohol intake. Completely replicating the advanced stages of alcohol-induced disorders observed in humans may be impractical. Third, this study focused on assessing alterations in the oral microbiota of rats with periodontitis prompted by continued alcohol consumption, rather than investigating the oral microbiota of rats with chronic alcohol consumption and periodontitis together. Future studies should evaluate changes in the oral microbiota induced by chronic alcohol consumption and periodontitis individually as well as conduct a horizontal comparison. Finally, the study employed functional prediction to explore potential mechanisms underlying the influence of chronic consumption of alcohol on periodontitis. However, the precise processes remain unclear and warrant further investigation.

In conclusion, this study provides novel insights into the changes in the oral microbiota of rats with periodontitis exacerbated by chronic alcohol consumption. Variations in the composition of oral microbiota were observed between the two groups, with an increase in

certain periodontal pathogenic bacteria. Chronic alcohol consumption may impact periodontitis by altering the carbohydrate metabolism function of the oral microbiota.

## CONCLUSIONS

Chronic alcohol consumption did not affect the diversity of the oral microbiota in rats with periodontitis. However, it did induce notable changes in the composition and functional characteristics of the oral microbiota. These alterations may contribute to the exacerbation of chronic periodontitis associated with chronic alcohol consumption, although the precise mechanism remains unknown. The observed shifts in composition and functional attributes of the oral microbiota provide valuable insights into the potential implications of chronic alcohol consumption for periodontal health.

### Funding

This research was supported by grants from the Clinical Medicine Talents Training Program of Hebei Provincial Government (361029) and the Innovative Experiment Program for College Students of Hebei Medical University. The funders had no role in study design, data collection and analysis, decision to publish, or preparation of the manuscript.

### Grant Disclosures

The following grant information was disclosed by the authors:
Clinical Medicine Talents Training Program of Hebei Provincial Government: 361029.
Innovative Experiment Program for College Students of Hebei Medical University.

### Competing Interests

The authors declare that they have no competing interests.

### Author Contributions

- Zirui Zhao conceived and designed the experiments, performed the experiments, analyzed the data, prepared figures and/or tables, authored or reviewed drafts of the article, and approved the final draft.
- Xiao Zhang performed the experiments, analyzed the data, prepared figures and/or tables, and approved the final draft.
- Wanqing Zhao performed the experiments, prepared figures and/or tables, and approved the final draft.
- Jianing Wang performed the experiments, prepared figures and/or tables, and approved the final draft.
- Yanhui Peng performed the experiments, prepared figures and/or tables, and approved the final draft.
- Xuanning Liu performed the experiments, prepared figures and/or tables, and approved the final draft.

- Na Liu conceived and designed the experiments, authored or reviewed drafts of the article, and approved the final draft.
- Qing Liu conceived and designed the experiments, authored or reviewed drafts of the article, and approved the final draft.

## Animal Ethics

The following information was supplied relating to ethical approvals (*i.e.*, approving body and any reference numbers):

This study was approved by the Ethics Committee of the Hospital of Stomatology Hebei Medical University, approval number: [2022]001.

## DNA Deposition

The following information was supplied regarding the deposition of DNA sequences:

The sequences described here are accessible *via* the NCBI Sequence Read Archive under the accession number PRJNA1087274.

## Data Availability

The raw measurements are available in the Supplemental Files 1 and 2.

## Supplemental Information

Supplemental information for this article can be found online at http://dx.doi.org/10.7717/peerj.17795#supplemental-information.

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
