# Peer review of "Effect of chronic alcohol consumption on oral microbiota in rats with periodontitis"

_PeerJ, doi:10.7717/peerj.17795_

## Round 0.1 · original submission · Minor Revisions

Dear authors,

The study entitled “Effect of chronic alcohol consumption on oral microbiota in rats with periodontitis” demonstrated excellent findings using an appropriate methodological approach. However, some important points must be clarified in the manuscript. Your article has great potential for publication on PeerJ, but the reviewers have requested substantial changes to be made, mainly in methodology and discussion sessions.

·

Basic reporting

I am not a native English speaker and I am not in a position to fully judge the professional quality of the English language, but I have noticed some misspelled words or sentences that are ambiguous. So, the quality of the English language must be rechecked.
The references are relatively recent and cover hypotheses, observations and discussions
the structure of the article is well composed. The figures and tables support the hypotheses.
Figure 2 and 3 must also be explained on the figure with arrows (not only in the text)
The results obtained through the bioinformatics analysis support the hypotheses of the study

Experimental design

The study falls within the aims and scope of the PeerJ journal, being a translational experimental medicine research
The article does not mention the accommodation and maintenance conditions of the animals. I suggest the authors to follow the ARRIVE guide (Animal Research: Reporting of In Vivo Experiments) see Kilkenny C, Browne W, Cuthill IC, Emerson M, Alman DG . NC3Rs Reporting Guidelines Working Group. Animal research: reporting in vivo experiments: the ARRIVE guidelines. J Gene Med. 2010; 12, 561–563. 10.1002/jgm.1473
My observations and questions are:
-how long did the period of adaptation to alcohol last?
- what it means ,,followed by maintenance feeding with the 30% alcohol solution for 10 weeks until the end of the experiment, during which they were forbidden to water”? - is unclear
- how was it determined how much alcohol a rat drinks per day? is there a comparison with water consumption?
-when did the induction of peri-implantitis begin-at what time period from the beginning of the study?
also related to the design of the experiment, another question would be:
were samples cultured on specific culture media for the groups of bacteria in the oral cavity?

Validity of the findings

The studies were not done in duplicate, so we do not know if there is replication.
But the bioinformatics analysis, which was done professionally and carefully, support the results and conclusions.

Additional comments

I also noticed some small issues in the text, which need to be solved
-not everywhere the genera of bacteria are written in italics
-there are abbreviations where the words are not mentioned in full first

Reviewer 2 ·

Basic reporting

The manuscript is well written, unambiguous and it has enough literature references.

Minor comments:
line 42: write chronic in lowercase - "In rats, Chronic alcohol consumption"
line 58: I suggest to replace "noncommunicable" with non-transmissible
line 99: write administration in lower case - "The alcohol Administration was based"
line 111: I suggest to add "we" before the verb "examine" - "At the end of the tenth week, examine the periodontal"

Experimental design

The methods are described in sufficient detail so that the experiment is replicable.

Did you follow the ARRIVE protocol for your experiment? Animal experimentation requires, in addition to approval by committees, following standardized experimental protocols. Animal Research: Reporting of In Vivo Experiments (ARRIVE) is a set of guidelines that was developed to improve the quality of reporting of experiments involving animals. If you use the ARRIVE guidelines I suggest to mention it in your manuscript.
Also, I recommend to mention the limitations of the study, especially the limitations related to the animal model used.

Validity of the findings

No comment.

Additional comments

No comment.

---

## Round 0.2 · accepted · Accept

Dear Author,

Congratulations! After your diligent work addressing the reviewers' comments, I am pleased to inform you that your manuscript has been accepted for publication in PeerJ. This version is more concise and formal, enhancing clarity and flow.

·

Basic reporting

The Authors have addressed all the remarks. The article can be published.

Experimental design

The Authors have adressed all the remarks. The article can be published.

Validity of the findings

The Authors have addressed all the remarks. The article can be published.

Additional comments

The Authors have addressed all the remarks. The article can be published.

Reviewer 2 ·

Basic reporting

The authors have adequately addressed all my comments and revised it accordingly.
The manuscript is significantly improved upon the readability and clarity of the manuscript. It is well-structured, designed, and referenced. Therefore, I have no further comments.

Experimental design

The authors have adequately addressed all my comments and revised it accordingly. Therefore, I have no further comments.

Validity of the findings

I have no further comments.

Additional comments

I appreciate the point-by-point response to my review and I want to congratulate the authors for their hard work.